# Review of Language Models for Survival Analysis

**Vincent Jeanselme**[1, 2, *]**, Nikita Agarwal**[2]**, Chen Wang**[2]

[1]University of Cambridge, MRC Biostatistics Unit
[2]Mayo Clinic

## Abstract

By learning statistical relations between words, Large Language Models (LLMs) have presented the capacity to capture meaningful representations for tasks beyond the ones they were trained for. LLMs' widespread accessibility and flexibility have attracted interest among medical practitioners, leading to extensive exploration of their utility in medical prognostic and diagnostic applications. Our work reviews LLMs' use for survival analysis, a statistical tool for estimating the time to an event of interest and, consequently, medical risk. We propose a classification of LLMs' modelling strategies and adaptations to survival analysis, detailing their limitations and strengths. Due to the absence of standardised guidelines in the literature, we introduce a framework to assess the efficacy of diverse LLM strategies for survival analysis.

## Introduction

In recent years, the advent of LLMs has sparked significant interest within the medical community (Bommasani et al. 2021; Garg et al. 2023; Li 2023; Thirunavukarasu et al. 2023; Wang et al. 2023a; Yang et al. 2022), with applications ranging from medical training (Lee 2023) and triaging (Levine et al. 2023) to drug discovery (Chakraborty, Bhattacharya, and Lee 2023).

LLMs empower practitioners to extract valuable insights from unstructured medical data, providing a potential tool for adverse events' diagnosis and prediction (Huang, Altosaar, and Ranganath 2019). Particularly, we explore how LLMs could be used for survival analysis, often used to quantify the risk of occurrence of an event of interest at different horizons but traditionally relying on structured covariates, e.g., 5-year risk of cardiovascular disease from vital sign and lifestyle measurements.

Our literature review identifies two ways LLMs can improve survival analysis and impact medical practice. First, LLMs offer a novel set of tools to alleviate the prohibitive cost and associated time of obtaining structured data, reducing the use of existing risk models (De Lusignan 2005; Hobbs et al. 2010; Jonnagaddala et al. 2015; Müller-Riemenschneider et al. 2010; Perera et al. 2017). Second,

LLMs facilitate the development of models directly from unstructured data, potentially improving predictions based on structured data alone.

**Contributions.** Recent reviews, such as the one by Hoekstra, Hurst, and Tummers, have delved into natural language processing for survival analysis. However, the evolving landscape of LLMs necessitates a detailed exploration of novel strategies for survival analysis and an assessment of their limitations. Particularly, this review contributes in three main ways: (i) classifying LLMs modelling approaches, (ii) reviewing their adaptation for survival analysis, and (iii) offering an open-source framework on Github[1] to evaluate these strategies. By inviting practitioners to compare these strategies on diverse datasets, we aim to develop evidence-based recommendations for applying LLMs in survival analysis tasks.

## LLM-based modelling

This section summarises modelling strategies using LLMs proposed in the literature, both as neural networks and interactive language tools through their generative capabilities. Figure 1 visually summarises the identified strategies, illustrating the transition from model-specific to model-agnostic learning. Before delving into these strategies, let's first establish what a LLM entails.

**Definition 1 (Large Language Model)** *A Large Language Model is a type of neural network designed to uncover statistical relationships between tokens, capturing informative representations. The term 'Large' emphasises the number of parameters, the amount of training data, and the computational resources required for training these models.*

### Embedding: *Leveraging deterministic representations*

**Description.** In scenarios with limited labelled data, a possible strategy involves deploying a pre-trained LLM to extract embeddings from unstructured data. This approach relies on the inherent capacity of LLM to represent unstructured data without additional training. First, an embedding – a vector of the values associated with a subset of LLMs'

---

[1]https://github.com/Jeanselme/LLM-For-Survival-Analysis

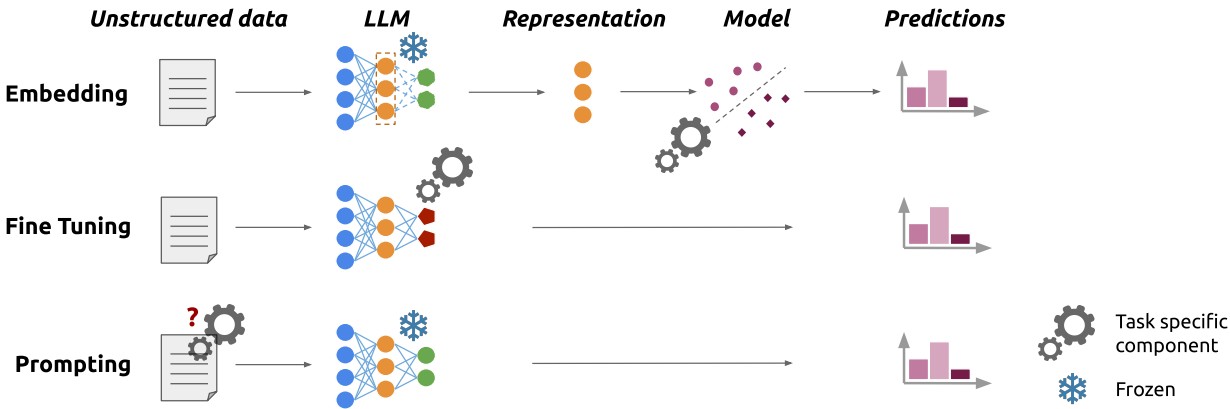

Figure 1: Overview of modelling approaches using Large Language Models.

inner nodes – is extracted and then used as inputs for a task-specific model.

**Strengths.** A critical advantage of this approach is its reliance on a pre-trained LLM, reducing the need for labelled data solely for training the task-specific model.

**Limitations.** This step-wise approach may result in suboptimal performance if the extracted representation fails to capture informative nuances from the domain-specific unstructured data. To address this limitation, multiple models have been trained on substantial amounts of domain-specific data to capture more relevant embeddings (Huang, Altosaar, and Ranganath 2019; Lee et al. 2020; Li et al. 2020; Lin et al. 2023; Moor et al. 2023). In the following, we refer to these models as domain-specific LLMs in contrast to general purpose LLMs.

### Fine-tuning: *Adjusting LLMs for the task*

**Description.** Fine-tuning entails adjusting the weights of a LLM using domain-specific labelled data to refine its representation for the task at hand. To accommodate the associated labels, one modifies the LLM's architecture, typically by replacing the last layer(s) of the LLM, and backpropagates the task-specific loss through the altered architecture.

**Strengths.** Fine-tuning presents improved performance with less data compared to training from scratch (Micheli, d'Hoffschmidt, and Fleuret 2020), as it takes advantages of the LLM's already learnt structures, while remaining more flexible than relying on fixed embeddings.

**Limitations.** The method still necessitates substantial amounts of data (Brown et al. 2020) and computation, potentially limiting its applicability in scenarios with small medical cohorts. Additionally, there is an inherent risk of overspecialisation, leading to a decrease in out-of-distribution generalisation (McCoy, Pavlick, and Linzen 2019). Gu et al. argue that, with sufficient data, training a model from scratch may outperform a fine-tuned model trained on a more general vocabulary. This observation emphasises the trade-offs associated with fine-tuning and data availability.

### Prompting: *Querying in natural language*

**Description.** LLMs are often trained as generative models, such as Generative Pre-trained Transformers (GPTs) (Brown et al. 2020). Relying on this property, prompting involves querying the LLM in natural language[2] and using the generated response as an estimate for the desired outcome.

**Strengths.** While the other approaches for using language models have long been established in machine learning, the concept of prompting has recently gained attraction (Brown et al. 2020). This interest stems from the strategy's general purpose, absence of training, and interactive nature.

**Limitations.** Prompting is not without challenges. First, it assumes that the user's articulation of the task and the model's ability to discern textual statistical correlations result in accurate predictions. *Assumption that needs to be carefully evaluated.* As the task-specific component is no longer data-driven, but specified by user[3], performances are highly dependent on the prompt (Mishra et al. 2021; Wang et al. 2023b). Second, estimating the probability distribution of the generated prediction, either through the returned probability vector or by sampling from the LLM, is crucial for non-deterministic models. It is important to remember than using a single prediction may not adequately represent the most likely one. Third, if the LLMs have not encountered similar data, there is no guarantee they can handle the type of data or task that the user presents, increasing the risk of inaccuracies. These limitations underscore the need for careful consideration and evaluation when adopting prompting strategies with LLMs.

---

[2]Refer to (Liu et al. 2023) for prompting strategies.

[3](Pryzant et al. 2023) proposes a textual gradient descent to optimise the prompt for best performance.

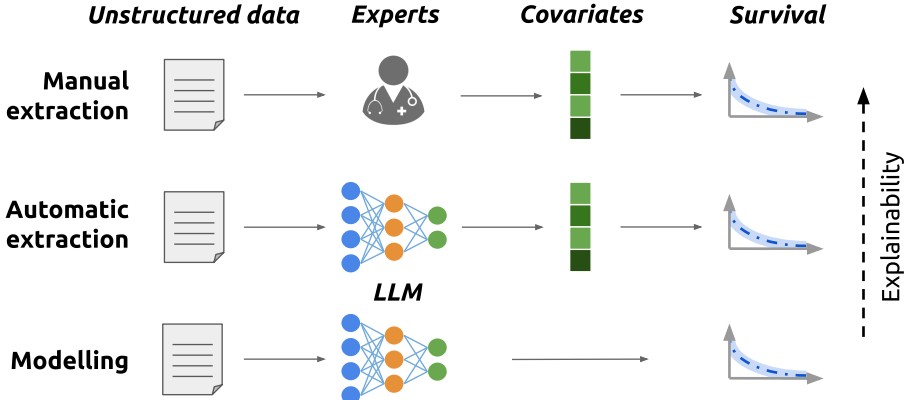

Figure 2: From medical notes to survival prediction, LLMs can be used for automatic covariates extraction or direct prediction.

## LLMs' adaptation for survival analysis.

Consider a dataset of the form $(u_i, x_i, t_i, d_i)$, with $u_i$, the unstructured data associated with a patient $i$, $x_i$, its structured covariates, $d_i$, the event indicator ($d_i = 0$ for patients who did not experience the event of interest over the study, known as censored patients, and $d_i = 1$ for those who did) and $t_i$, the associated time.

Estimating a patient's risk consists of accurately estimating the probability of observing the event of interest before a time $t$. This quantity is known as the survival function (Collett 2023; Klein, Moeschberger et al. 2003) and defined as:

$$S(t) = \mathbb{P}(T < t)$$

with $T$, the random variable associated with the event time.

In medical research, practitioners aim to estimate how the structured covariates $x_i$ influence $S$ to recommend treatment and inform medical decisions. Due to their complex statistical analysis and interpretation, unstructured data $u_i$ have often been discarded from this analysis.

Through our literature review (outlined in the Appendix), we identified two key purposes in using LLMs: (i) improving the adoption of existing models by lowering the time and cost associated with the extraction of the covariates $x_i$ that limit the use of existing models (De Lusignan 2005; Hobbs et al. 2010; Müller-Riemenschneider et al. 2010; Perera et al. 2017), and (ii) leveraging information in patients' unstructured data to model the outcome of interest. Consequently, we classify existing works as (i) automatic extraction and (ii) survival modelling.

## Automatic extraction: *from unstructured to structured data*

To mitigate manual labour (Bush et al. 2017) and reduce costs, LLMs emerge as valuable tools for covariates extraction from unstructured data. Extracted covariates can subsequently be used for evaluating, or developing, survival models. Schematically, the step-wise pipeline is as follows:

$$u_i \xrightarrow{\text{LLM}} x_i \xrightarrow{\text{Survival Model}} t_i$$

In the following, we describe how to use the previous LLM strategies for automatic extraction, and reference existing works in the literature.

**Embedding.** After embedding the unstructured data through an LLM, automatic extraction becomes a traditional classification or regression problem modelling $x_i$ given the embedding $\tilde{u}_i$.

**Fine-tuning.** After altering an LLM's architecture to contain a final classification layer, the model is fine-tuned using pairs of $(u_i, x_i)$. For instance, (Khurshid et al. 2022) propose to impute missing values in electronic health records using nurses' notes by fine-tuning a BERT (Devlin et al. 2018) and an alternative architecture previously trained on medical data and discharge summaries. Similarly, (Hsu et al. 2023) impute stroke features from imaging notes, after fine-tuning ClinicalBERT (Huang, Altosaar, and Ranganath 2019) on the imaging notes (creating a domain-specific LLM) and then, further fine-tuning the altered architecture on 200 annotated pairs of unstructured notes and structured features.

**Prompting.** Using a generative model, one can query the model as, for example,: "*For the patient described through the following report [$u_i$], extract the patient's: age = [?], sex = [?], diabetes status = [?].*". (Agrawal et al. 2022; Truhn et al. 2023) introduce diverse prompting strategies to extract clinical concepts from notes. (Gero et al. 2023; Wei et al. 2023) introduce further enhancements through self-verification mechanisms, i.e., iterative querying of the LLM.

**Literature's recommendations.** Automatic extraction of medical concepts presents a long history (Caccamisi et al. 2020; Cowie and Lehnert 1996; Jonnagaddala et al. 2015; Meystre et al. 2008; Wang et al. 2018; Weissman et al. 2018) as it presents the attractive properties of (i) independence from the downstream task, (ii) allowing use of well-known and interpretable statistical tools both at development and deployment.

While (Khurshid et al. 2022) shows the superiority of LLMs over standard rule-based approaches, the literature does not offer conclusive recommendations on which of the

three strategies to prefer. When using the fine-tuning strategy, (Hsu et al. 2023; Khurshid et al. 2022) recommend the use of domain-specific LLMs over more general architectures for improved extraction. (Gutierrez et al. 2022) echoes this recommendation and further demonstrates the superiority of fine-tuning BERT architectures over prompting GPT-3 for biomedical concept extraction from medical abstracts. However, for clinical notes, (Agrawal et al. 2022) concludes that prompting GPT-3 outperforms fine-tuned BERT architectures on treatment extraction.

These results highlight the complexity of choosing the best strategies due to the data type, data size, task, strategies, LLMs and training implementations.

## Survival modelling: *from unstructured data to risk estimate*

When predictive performance is the primary goal, a direct approach involves modelling risk from unstructured data, schematically summarised as:

$$u_i(, x_i) \xrightarrow{\text{LLM}} t_i$$

**Embedding.** The embedding strategy employs LLMs' inner representations as inputs for a survival model, trained independently. For example, (Kim et al. 2021; Lee et al. 2021; Likith, Begam, and Shashikant 2022) use LLMs to extract embeddings from MRI, radiology and clinical reports using BERT-based architectures. (Kim et al. 2021; Lee et al. 2021) further use a Long Short Term Memory (Hochreiter and Schmidhuber 1997) to agglomerate longitudinal reports into a single representation. Then, the authors use a Cox model (Cox 1972) to predict the risk for different events. Alternatively, one can use traditional classification models to predict binarised outcomes such as cancer recurrence (Kaka et al. 2022), death (Huang, Altosaar, and Ranganath 2019; Li et al. 2023; Wang et al. 2022) or chronic cough (Luo et al. 2021).

**Fine-tuning.** By appending a last layer to the LLM with one node per outcome of interest, one can learn a fine-tuned representation for the task at hand. (Huang, Altosaar, and Ranganath 2019; Jiang et al. 2023; Luo et al. 2021; Lin et al. 2023; Mugisha and Paik 2020; Munoz-Farre, Rose, and Cakiroglu 2022) append a last layer to the BERT architecture (or a domain-specific version) for binary risk estimate. To account for censoring, (Zhao et al. 2021) fine-tunes a BERT architecture with a final node used as the relative risk in a Cox regression model and use the relative log-likelihood to train the model.

**Prompting.** Discretisation of the survival outcome, i.e. determining whether the patient experiences the outcome of interest within a given time horizon, offers a straightforward prompting strategy. For instance, (Han et al. 2023a) query ChatGPT with "*Estimate the risk (in percentages) of developing a cardiovascular disease within 10 years for the person below: [$u_i$]?*" using semi-synthetic notes obtained by describing structured data from UK Biobank (Sudlow et al. 2015) and KoGES (Kim, Han, and Group 2017). Despite ignoring the model's uncertainty in the generated response, the

analysis demonstrates that the larger GPT-4 improves performance compared to smaller LLMs and performs similarly to traditional risk scores.

**Literature's recommendation.** Unstructured data may contain information that improves performance over manually extracted covariates (Mugisha and Paik 2020; Pandey et al. 2020). However, this conclusion is dependent on the approach and model used.

When using the embedding strategy, (Lin et al. 2021; Philonenko, Kokh, and Blinov 2023) report improved performance when relying on LLMs' representations of unstructured data compared to structured data alone. However, LLMs perform similarly to traditional word frequency representations in (Klang et al. 2022) or manually extracted features in (Fanconi, van Buchem, and Hernandez-Boussard 2023). Note that these two previous works only consider general-purpose LLM that may not be adapted to the considered unstructured data. Critically, (Lee et al. 2021)'s analysis demonstrates the superiority of domain-specific LLMs' embeddings over manually extracted features and general-purpose LLMs. (Wang et al. 2022) reaches similar conclusions with improved binary predictions using clinical notes embedded through ClinicalBERT compared to Word2Vec (Mikolov et al. 2013). This discussion comes with nuances as the efficacy of domain-specific LLMs may be data-dependent, as noted by (Kaka et al. 2022) with a limited improvement of ClinicalBERT over BERT on medical records.

In the context of fine-tuning, studies by (Huang, Altosaar, and Ranganath 2019; Jiang et al. 2023; Mugisha and Paik 2020) show that fine-tuning a domain-specific model to predict risk outperforms fine-tuning a more general model or using bag-of-word baselines. Importantly, (Jiang et al. 2023) empirically demonstrates that domain-specific models present better performance with smaller amounts of data.

For prompting, the literature focuses on demonstrating the model's generative capacity to predict outcomes, and has not explored the superiority of domain-specific LLMs over more general ones for survival task. Critically, the discussion revolves around the use of unstructured data and the choice of LLMs, often leaving out the question of which strategy should be preferred.

## Discussion

Our literature review highlights important considerations for (i) the development of survival models from unstructured data, (ii) their application in clinical practice, and (iii) LLMs' development.

### Survival modelling

In medical studies, patients often do not experience the event of interest over the study period. This central problem, known as censoring, is often ignored. For instance, many reviewed studies rely on outcomes' binarisation without censoring adjustment. Critically, ignoring censored patients biases time-to-event estimates (Turkson, Ayiah-Mensah, and Nimoh 2021), as censored patients remained event-free until they left the study. When explicitly considered, reviewed

works rely on the Cox model, whose proportionality assumptions may not hold in medical data (Stensrud and Hernán 2020).

Our work calls practitioners for careful consideration of time-to-event challenges, namely censoring and competing risks. Neural network approaches have tackled these challenges such as (Danks and Yau 2022; Jeanselme et al. 2023; Lee et al. 2018) and could be considered jointly with LLMs.

### Clinical actionability

The survival literature has focused on performance over actionability. While models' low accuracy is a barrier to adoption (Hobbs et al. 2010), the critical connection between risk and medical recommendation is even more critical (Hobbs et al. 2010). The focus should shift from performance alone to survival models' actionability as discussed in (Jeong et al. 2024).

In this context, the direct prediction of risk from unstructured notes appears disconnected from medical practice, unless one can derive medical recommendations from them. The automatic extraction strategy may allow the development of traditional risk models in which exposure can be connected to outcomes. However, we must question the hypothesis that automatic evaluation would improve risk models' deployment by reducing the cost of obtaining structured data. Critically, does the computational cost of evaluating risk on a larger population with potential machine errors resulting in additional tests, actually lower the cost and improve patients' outcomes compared with current practice?

Collaborations to study these multiple challenges are crucial to translate the development of new models into improved use and care.

### LLMs' development

Despite the prevalence of censoring in medical studies (Lesko et al. 2018) and methodological advances in survival analysis, censoring has received little attention in the development of LLMs. While the current focus on medical LLMs (Huang, Altosaar, and Ranganath 2019; Lee et al. 2020; Li et al. 2020; Lin et al. 2023; Moor et al. 2023; Yang et al. 2023) recognises the need to enhance representation by learning from large amounts of domain-specific data and available labels, the challenges posed by unobserved outcomes and data imbalances associated with censoring are often overlooked.

Critiques highlight the disconnect between LLMs approaches and their relevance to medicine (Shah, Entwistle, and Pfeffer 2023; Wornow et al. 2023), calling for using more medical data in LLMs' development. We would like to extend the conversation by emphasising the necessity of accounting, not only for domain-specific data but for domain-specific challenges. For instance, addressing the often-overlooked issue of censoring is critical for medical relevance. Despite the recent development of foundational models for medical predictions, few mention the problem of censoring. Only Steinberg et al. proposes a foundational model to predict the time to the next events and demonstrates the superiority of the foundational model over task-specific ones in the context of electronic health records.

## Proposed Evaluation Framework

Our review highlights the lack of standardised evaluation frameworks to compare the introduced LLMs' strategies. Studies employ different datasets, tasks, approaches, models and implementations, limiting possible comparison. Further, the over-reliance on the MIMIC (Johnson et al. 2016) dataset in both training domain-specific LLMs and modelling raises concerns about potential leakage and limits the generalizability of findings.

To obtain evidence-based recommendations on the use of LLMs for survival predictions, we introduce the following framework. This framework aims to fix the models and training pipeline to obtain comparable evidence across datasets. To this end, we provide an implementation on GitHub[4] with a tutorial to tailor the pipeline to practitioners' datasets. We invite practitioners to evaluate this framework on their data and share their findings to guide recommendations.

In the following, we detail our framework with an example on the publicly available Cancer Genome Atlas (TCGA) dataset (Tomczak, Czerwińska, and Wiznerowicz 2015), and the associated pathology reports (Kefeli and Tatonetti 2023) available on Github[5]. For each patient, a report ($u_i$), manually extracted demographics and cancer stage ($x_i$), and survival or censoring times ($t_i, d_i$) are recorded.

### Training

As multiple centres provided data to the TCGA study, we propose a 3-fold cross-validation stratified by hospitals to quantify the different strategies' generalisability to new institutions where reporting guidelines may differ. As all experimental settings may not allow this evaluation, we additionally implement a standard 3-fold cross-validation. We rely on open-source models from HuggingFace (Wolf et al. 2019) to ensure reproducible results while maintaining data privacy.

**Automatic extraction.** The following describes the three LLMs approaches for the extraction of the structured data $x_i$ from the unstructured report $u_i$.

*Embedding.* To embed the unstructured data, we use encoder-decoder architectures, more amenable to this task. Specifically, we rely on BERT (Devlin et al. 2018) as a general-purpose LLM and a domain-specific LLM: ClinicalBERT (Huang, Altosaar, and Ranganath 2019) which has been fine-tuned on PubMed publications and then MIMIC clinical notes. By considering both LLMs, we aim to quantify the gain of using domain-specific LLMs. We save the extracted embeddings for analysing both automatic extraction and the survival modelling strategy. For extraction of the structured data from the embedding, we use a multi-layer perceptron with one output per hot-encoded covariates trained for 100 epochs[6] with early stopping criterion using an Adam optimiser to minimise the cross-entropy loss.

---

[4]https://github.com/Jeanselme/LLM-For-Survival-Analysis

[5]https://github.com/tatonetti-lab/tcga-path-reports

[6]Note that we allow a larger number of epochs for the embedding strategies as a larger number of parameters need to be learnt from scratch.

*Fine-tuning.* This approach relies on the same LLMs concatenated with a one-layer perceptron with one node per covariate. The full architecture is trained for 10 epochs using an Adam optimiser to minimise the cross-entropy loss.

*Prompting.* For prompting, we rely on open-source generative LLMs: Llama 7b (Touvron et al. 2023) and MedAlpaca (Han et al. 2023b) as a domain-specific LLM. For automatic extraction, we iteratively query: "Context: Pathology report $u_i$ Question: Based on the provided pathology report, what is the covariate (possible values: possible covariate values or range)? Please provide your answer as one of these values, without any additional text or explanations. Answer:". To ensure reproducibility (and self-consistency), we reduce the temperature to ensure a deterministic generation. Note that this results in considering only the most likely generated sequence but does not account for the potential uncertainty associated with the prompt.

**Survival Modelling.** The previous approaches lead to the extraction of structured data. The following presents how we model the survival outcome from these covariates and from the unstructured data. Specifically, we discretize the survival outcome into 4 time intervals: death within $[0-1]$ year, $[1-3]$, $[3-5]$ and more than 5 years after diagnosis, and use the log likelihood for training as in DeepHit (Lee et al. 2018).

*Covariates.* From the covariates, we train a neural network consisting of 3 hidden layers with 50 nodes with a final layer with one output per time interval. We maximise the following log-likelihood over 100 iterations using an Adam optimiser:

$$\mathcal{L}_{\text{DeepHit}} := \sum_{i,d_i=1} \log(N_{t_i}(x_i)) + \sum_{i,d_i=0} \log(1 - N_{\leq t_i}(x_i)),$$

with $N_t(x)$, the neural network's output corresponding to the probability of having the event in the time interval containing $t$ given the covariate $x$.

*Embedding.* The same modelling than described for *Covariates* is used when using embeddings.

*Fine-tuning.* Similarly, after aggregating a one-layer perceptron with one node per time discretisation to a BERT or ClinicalBERT model, the architecture is trained using the previous log-likelihood. We refer to this model as LLMHit as an extension of the traditional DeepHit to LLM.

*Prompting.* To predict patient's survival, we adapt (Han et al. 2023a)'s prompting approach to the LLMs' formatting and query the models with the following "Context: Pathology report $u_i$ Question: Based on the provided pathology report, what is the estimated probability (between 0 and 1) that the patient will die within the next horizon years? Please provide your answer as a single decimal number rounded to two decimal places, without any additional text or explanations. Answer:". We repeat this prompt for each time horizon.

## Evaluation

To measure the quality of the different approaches, we rely on two common survival metrics: the C-Index (Antolini, Boracchi, and Biganzoli 2005) measuring discriminative performance and the Brier Score (Graf et al. 1999) quantifying calibration integrated over the three considered time horizons after diagnosis. Additionally, we compute the mean squared error for the quality of the automatic extraction

## Conclusion

This paper presents a classification of the different strategies for using LLMs for survival analysis and highlights the current lack of recommendations in this field. As a remedy, we propose an evaluation framework to facilitate comparisons between LLMs' strategies and settings. We invite practitioners to evaluate these strategies and contribute to this framework to guide the development of future time-to-event models to together develop evidence-based answers to the question: "*Which LLM strategies should be preferred, for what type of data and research question?*".

## Ethical statement

While this work focuses on the descriptions of the different strategies used for time-to-event modelling, we would like to echo some critical risks of these approaches (Bender et al. 2021). The reliance on unstructured data raises the concern of what these modalities embed. Beyond capturing a patient's health, medical notes can reflect practitioner's fatigue (Hsu, Obermeyer, and Tan 2023), and missing covariates may present biases (Jeanselme et al. 2022). In a field marked by historical inequities, LLMs may learn and repeat these inequities (Navigli, Conia, and Ross 2023). Consequently, we echo (Wang, Zhao, and Petzold 2023) and call for caution when employing these models, particularly as they become less amenable to corrections, potentially leading to ever-harmful consequences.

## Acknowledgement

This research is supported by the Eric and Wendy Schmidt Fund for AI Research and Innovation, and the Mayo Clinic Center for Individualized Medicine.

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

## Literature Review

This semi-systematic review was conducted using Google Scholar with the prompt *"survival analysis" OR "time-to-event" AND "language model" AND "medicine" OR "healthcare"* for publications between 2018 (chosen as it marks the publication of the seminal work by Devlin et al.) and 2024 (excluded). This query led to 335 publications containing these terms in their title or abstract. We sub-selected papers with at least an experiment relying on medical text modality and a survival outcome.