# OpenReview forum: "Review of Language Models for Survival Analysis"
_AAAI.org/2024/Spring_Symposium_Series/Clinical_FMs — AAAI 2024 SSS on Clinical FMs_

### Official Review · Reviewer_ZyZX · 2024-02-13
**An important effort in need of a few improvements**

**Rating:** 5
**Confidence:** 4

**Review:**

The work addresses an important problem - the current lack of a meaningful comparison of methods for application of LLMs to process unstructured data for clinical survival analysis. It is very well-written, organized, and enjoyable to read. I am not well-versed in this subfield, but it appears to be reasonably comprehensive. It provides a framework for addressing the issues brought up in the review.

There is also room for improvement. The descriptions of many works lack details and are hard to draw conclusions from. At several points recommendations are given, but they are not very clear or specific. If the point is to highlight that prior work does not support clear recommendations, it would be clearer to state this, as the current structure makes it appear that the intent is for the authors to give methodological recommendations based on the reviewed work. A large part of the contribution is a common evaluation framework, which addresses the gaps in the field highlighted by the review; however, this is largely relegated to the appendix, not described in much detail, and not justified clearly in its design choices. I cannot view the current state of the project as the link is hidden for anonymity.

Also, I would point out that the strategy used for literature search seems brittle. For example, the search includes titles with either "survival analysis" or "time-to-event" AND "medicine" or "healthcare". Ironically, this submission itself meets neither of these criteria. This suggests both the need for a more robust search strategy and possibly the need for a more descriptive title for this submission. It would also be helpful if it was clearer from the title that this is a review.

Questions and suggestions for the authors:
1. In section "Fine-tuning: Adjusting LLMs for the task - Limitations": are there any conclusions to be drawn from this? The findings seem contradictory.
2. Prompting: Querying in natural language - Strengths. Can you elaborate? The main point given is that these methods are the most novel, which is not a strength per se.
3. I found the following statement confusing: "However, multiple risk scores are rarely evaluated due to the prohibitive cost of extracting the required covariates x_i, which are often present in patients’ unstructured health records." What are the multiple risk scores being referenced here?
4. You encourage researchers to "compare LLMs strategies on private sources using our implementation." Can you clarify how this works? Is the data private, and if so, how do other researchers use and present it?
5. What is the current state of the GitHub? Can it be used yet? Can you provide any results for the methods that are implemented so far? I also understand that the intent is partially to use this conference to gather feedback for its improvement.

With some clarification on the nature of the provided evaluation framework, I think the paper meets the threshold for acceptance in its current state, but I hope the authors will take this feedback into account, as it could be a stronger paper without too much effort.

---

### Official Review · Reviewer_narw · 2024-02-20
**The graphs seems not well orgnized, could you please double theck the graphs**

**Rating:** 6
**Confidence:** 3

**Review:**

The paper emphasizes the potential of LLMs in extracting insights from unstructured medical data for survival analysis and risk estimation, proposing a comprehensive approach that incorporates embedding, fine-tuning, and prompting strategies. It also highlights the importance of cross-validation techniques in assessing the generalizability of these methods across different medical institutions.
The graphs seems not well orgnized, could you please double theck the graphs. Besides, the recommendation section seems fragmented, could it be reorganized to summarize it in the conclusion?

---

### Official Review · Reviewer_qHUF · 2024-02-21
**Leveraging Language Models for Risk Estimation in Medical Prognostic and Diagnostic Applications**

**Rating:** 7
**Confidence:** 3

**Review:**

The paper discusses leveraging Large Language Models (LLMs) for survival analysis in medical prognostic and diagnostic applications. It explores methodologies for estimating medical risk, with a focus on survival analysis, addressing the challenges posed by censoring in medical studies. The document reviews current LLM strategies for survival analysis, detailing their limitations and strengths, and proposes an open-source implementation for comparing these strategies. It aims to develop evidence-based recommendations for the effective use of LLMs in estimating patient survival outcomes.

**Pros**
- Comprehensive Approach: The paper provides a thorough overview of using LLMs for survival analysis, covering various methodologies and their adaptations for this specific application.
- Practical Contributions: By offering an open-source implementation, the work facilitates practical experimentation and comparison of different LLM strategies for survival analysis.
- Addressing a Crucial Challenge: The paper tackles the significant challenge of censoring in survival analysis, proposing methods to improve modeling under such conditions.

**Cons**
- Generalization of Findings: The paper might benefit from a broader evaluation across different types of LLMs and datasets to ensure the findings' generalizability.
- Detailed Evaluation Framework: While it proposes an open-source implementation for comparison, the paper could provide a more standardized evaluation framework for assessing the performance of LLM strategies.